# Parental Self-Efficacy in Managing Pediatrics’ Medications and Treatments in Jordan: A Cross-Sectional Study

**DOI:** 10.3390/healthcare13182280

**Published:** 2025-09-12

**Authors:** Abdallah Y. Naser, Hassan Al-Shehri

**Affiliations:** 1Department of Applied Pharmaceutical Sciences and Clinical Pharmacy, Faculty of Pharmacy, Isra University, Amman 11622, Jordan; abdallah.naser@iu.edu.jo; 2Department of Pediatrics, College of Medicine, Imam Mohammad Ibn Saud Islamic University (IMSIU), Riyadh 13317, Saudi Arabia

**Keywords:** children, medications, parent, self-efficacy, self-management

## Abstract

**Background:** Parents make vital decisions regarding their children’s health and safety. Poor parental self-efficacy is associated with unfavorable health outcomes among their children. This study aims to investigate parental self-efficacy in managing pediatric medications and treatments in Jordan. **Methods:** This is an online cross-sectional survey study that was conducted in Jordan between 20 April and 4 July 2025. Self-efficacy in managing medications and treatments for children was assessed utilizing a previously validated questionnaire, including healthcare information or decision-making, symptom identification or management, general treatment management, general healthcare navigation, and feeding management. Logistic regression analysis was performed to identify predictors of a higher level of self-efficacy. **Results:** A total of 597 parents were included in this study. The majority of parents reported high levels of confidence (self-efficacy) in managing various aspects of their child’s care. The highest proportion of parents indicated they were very confident in knowing when their child needs to visit a healthcare provider (35.2%) and in following their child’s diet or nutrition plan (36.9%). Very confident was the most selected response for knowing how to contact healthcare providers (38.4%) and scheduling an appointment (37.0%). Higher income was strongly linked to greater self-efficacy, with parents earning 1001–1500 Jordanian dinars (JOD) showing significantly higher odds (odds ratio (OR) = 4.44, 95% confidence interval (CI): 2.42–8.15, *p* < 0.001) compared to those earning less than 500 JOD. Parents working in medical fields also had higher odds (OR = 3.30, 95% CI: 1.69–6.45, *p* < 0.001) compared to those not working. Parents with 2–3 children (OR = 1.73, 95% CI: 1.00–3.00, *p* = 0.049) or 4–5 children (OR = 1.59, 95% CI: 1.05–3.63, *p* = 0.03) had greater odds of self-efficacy compared to those with one child. **Conclusions:** The majority of the parents in this study expressed strong self-efficacy in managing their child’s care, specifically in healthcare-related tasks. Higher self-efficacy was significantly associated with parents’ socioeconomic characteristics such as marital status, medical employment, income, insurance coverage, and number of children. At the same time, lower confidence levels and self-efficacy were observed among divorced parents. More support should be directed towards low-income families and parents who work outside the medical field to enhance their self-efficacy and ultimately the health outcomes of their children.

## 1. Introduction

Parental self-efficacy is defined as a parent’s perception of confidence in their capability to manage and accomplish parenting tasks effectively. Parental self-efficacy influences the behaviors of caregiving parents [1,2]. Consequently, it plays a crucial role in shaping the health behaviors of their child in early childhood [3,4]. Parents make vital decisions regarding their children’s safety measures, physical activity, type of food they will consume, and the quantity and quality of healthcare they will be able to obtain [5].

The greater self-efficacy adults and children have, the better they use coping strategies that promote health outcomes [6,7,8,9,10,11]. Acceptance and positive reframing are among the most effective coping strategies that can be adopted to enhance medication adherence and disease management in both the parents and their children [12]. Higher self-efficacy is associated with active coping engagement, which ultimately is associated with better pediatric health outcomes [12,13]. Follow-ups are essential to ensure and evaluate growth and nutrition, family connections, continued education, and medication compliance [11]. Parental involvement and interventions largely determine whether a child will adhere to medicine [14]. Considerations about the severity of the medical condition and parental self-efficacy will enhance adherence, while parental worries about the child’s non-adherence may decrease it [15].

Parental self-efficacy is also a determinant of a child’s social adaptation and school performance [16,17,18]. Additionally, a systematic review demonstrated correlations between various factors, including parental self-efficacy, perceived social support, household income, maternal depression, parenting stress, and maternal satisfaction with parenting [19]. Studies consistently demonstrate that the increased care burden is related to increased risk of the parents’ compromised health-related quality of life [20,21,22].

Pediatric populations are susceptible to a wide range of medical conditions that might warrant hospitalization and critical healthcare [23,24,25]. A recent study in Jordan found a moderate level of knowledge among parents concerning the use of antibiotics for their children [26]. Understanding parental self-efficacy in managing children is crucial for assessing their health and well-being, as well as for creating early and effective interventions to improve their overall well-being and health outcomes. Parental self-efficacy is influenced by sociodemographic characteristics and is associated with better disease control, safer medication administration, and higher levels of adherence [27,28,29]. However, such studies are lacking in Jordan. Thus, this study aims to investigate parental self-efficacy in managing pediatric medications and treatments in Jordan. The specific objectives were to identify the level of parental self-efficacy in managing medications and treatments in Jordan. Additionally, we aimed to identify predictors of a higher level of parental self-efficacy.

## 2. Methods

### 2.1. Study Design and Setting

This is an online cross-sectional survey study that was conducted in Jordan between 20 April and 4 July 2025.

### 2.2. Study Population and Sampling Procedure

Participation in the investigation was restricted to individuals who satisfied the inclusion criteria. The study’s participants were parents with children under the age of 18 who are currently living in Jordan. This investigation employed the convenience sampling technique to invite eligible participants to participate. Social media platforms (such as Facebook and WhatsApp) were employed to solicit research participants. The invitation letter, along with the survey link, was circulated using Qualtrics XM, software across Facebook and WhatsApp groups of Jordanians. The invitation letter highlighted the study’s aim and objectives along with the study’s inclusion criteria. Only participants who met the inclusion criteria were asked to participate.

The minimum required sample size was 383 parents, with a confidence interval of 95%, a statistical power of 80%, an assumed proportion of 0.5, and a margin of error of 5%. The following equation was used to estimate the sample size: *n* = *Z*^2^ × *P* (1 − *P*)/*d*^2^.

n is the sample size; Z is the Z-score; P is the assumed population proportion; and d is the margin of error.

### 2.3. Study Instrument

The questionnaire instrument was translated using the forward–backward translation technique and administered in the Arabic language and comprised two parts. The first part examined parents’ sociodemographic characteristics (parent age, gender, marital status, education level, income level, occupation status, insurance status, comorbidities history, number of children in the family, and the individual who takes care of the child). The second part consisted of 14 items on a 5-point Likert scale format that examined self-efficacy in managing medications and treatments for children utilizing a previously validated questionnaire by Foster et al. [27]. Participants’ response ranges from “not confident at all” (given a score of 1) to “extremely confident” (given a score of 5), with an estimated score that ranged between 14 and 70. The selected category was given a numeric value corresponding to the category number. The self-efficacy domain examined five domains, which are healthcare information or decision-making, symptom identification or management, general treatment management, general healthcare navigation, and feeding management. The higher the score, the higher the self-efficacy for the study participant.

### 2.4. Questionnaire Validity

The original questionnaire was developed based on targeted literature reviews that identified principal self-efficacy measures in managing diseases, focusing on the pediatric population. The questionnaire tool was checked and validated by 12 pediatricians. This was followed by interviewing participants who met the study inclusion criteria. The interview focused on assessing the questionnaire items in terms of their relevance to their child’s care and clearance.

For the current study, the validity of the questionnaire tool was examined through revising and assessing the questionnaire tool by an expert panel of pediatricians and clinical pharmacists. The expert panel confirmed the external validity of the tool. Additionally, a pilot study was conducted on 30 participants from the targeted study population. Similarly, the pilot study confirmed the clarity of the survey items. The internal reliability of the questionnaire tool was checked using Cronbach’s alpha test, which measured 0.957, indicating a very high level of internal consistency across the questionnaire items.

### 2.5. Ethical Approval and Informed Consent

The scientific research ethics committee of Isra University, Amman, Jordan, granted ethical approval for this study (SREC/25/04/141) (date: 20 April 2025). The World Medical Association (WMA) Declaration of Helsinki was adhered to during the execution of this study. The study questionnaire was completed only after informed consent was obtained from the study participants. The study participants were informed that completing the questionnaire tool was considered informed consent.

### 2.6. Data Analysis

Demographic variables, including age group, gender, education level, and occupation, were summarized using frequency and percentage. The total self-efficacy score, treated as a continuous variable, was described using the mean, standard deviation (SD). The normality of the score was evaluated using the Kolmogorov–Smirnov test. Independent sample *t*-tests were used to compare scores between two groups, such as gender, while one-way ANOVA was applied to assess differences across other demographic variables. When significant differences were found, Tukey’s post hoc test was used for pairwise comparisons. Logistic regression analysis was also performed to identify predictors of a higher level of self-efficacy, with scores dichotomized based on the mean at a cutoff of 46.43 (which is the mean self-efficacy score for the study sample). All analyses were carried out using SPSS version 29, (IBM, Chicago, IL, USA) with statistical significance at *p* < 0.05.

## 3. Results

The study included 597 parents, of whom 437 (73.4%) were female and 159 (26.6%) were male. The majority were married (n = 503, 84.3%), while 63 (10.6%) were divorced. The most common age groups were 30–34 years (n = 133, 22.3%) and 40–44 years (n = 102, 17.1%). In terms of education, 269 (45.1%) had a bachelor’s degree, 148 (24.8%) had a diploma, and 64 (10.7%) had a postgraduate degree. Regarding income, 271 (45.4%) earned less than 500 JOD and 209 (35.0%) earned 500–1000 JOD. Regarding employment status, 239 (40.0%) worked in medical fields, 211 (35.3%) worked outside medical fields, and 45 (7.5%) were retired. Most participants had health insurance (n = 411, 68.8%), and 173 (29.0%) reported comorbidities. In terms of caregiving, 324 (54.3%) were the primary caregivers, and 186 (31.2%) shared responsibility with another party, as shown in Table 1.

The majority of parents reported high levels of confidence (self-efficacy) in managing various aspects of their child’s care. The highest proportion of parents indicated that they were very confident in knowing when their child needs to visit a healthcare provider (n = 210, 35.2%) and in following their child’s diet or nutrition plan (n = 220, 36.9%). Similarly, very confident was the most selected response for knowing how to contact healthcare providers (n = 229, 38.4%) and scheduling an appointment (n = 221, 37.0%). In contrast, confidence was lower regarding setting up telemedicine visits, where the most common response was somewhat confident (n = 201, 33.7%), as shown in Figure 1.

The total mean of the score was 46.43 ± 11.67, with a maximum and minimum of 70–14. Married parents had a higher mean score (46.97 ± 11.37) compared to divorced parents (42.59 ± 10.66) (*p* = 0.01). Parents with a postgraduate education reported the highest self-efficacy (50.69 ± 10.52) versus those with less than a high school education (41.92 ± 13.13) (*p* = 0.001). Higher income was associated with greater self-efficacy, with parents earning over 1000 JOD showing notably higher scores (52.54 ± 8.45) than those earning less than 500 JOD (43.10 ± 12.07) (*p* = 0.001). Those working in medical fields reported higher self-efficacy (53.33 ± 9.51) compared to other occupational groups (*p* = 0.001). Additionally, primary caregivers reported higher scores (46.89 ± 10.81) than those who were not the main caregivers (43.57 ± 14.65) (*p* = 0.04), as shown in Table 2.

The logistic regression analysis model provided an acceptable fit to the data. The omnibus test of model coefficients was (χ^2^ (26) = 108.38, *p* < 0.001), and the findings of the Hosmer–Lemeshow goodness-of-fit test were non-significant (χ^2^ (8) = 8.69, *p* = 0.369).

Divorced parents had lower odds of having better self-efficacy compared to married parents (odds ratio (OR) = 0.49, 95% confidence interval (CI): 0.26–0.92, *p* = 0.028). Higher income was strongly linked to greater self-efficacy, with parents earning 1001–1500 JOD showing significantly higher odds (OR = 4.44, 95% CI: 2.42–8.15, *p* < 0.001) compared to those earning less than 500 JOD. Parents working in medical fields also had higher odds (OR = 3.30, 95% CI: 1.69–6.45, *p* < 0.001) compared to those not working. Additionally, having health insurance was associated with lower parental self-efficacy (OR = OR 0.50, 95% CI: 0.33–0.76, *p* = 0.001), and parents with 2–3 children (OR = 1.73, 95% CI: 1.00–3.00, *p* = 0.049) or 4–5 children (OR = 1.59, 95% CI: 1.05–3.63, *p* = 0.03) had greater odds of self-efficacy compared to those with one child, as shown in Table 3.

## 4. Discussion

In this study, the majority of parents reported high levels of confidence in managing various aspects of their child’s care. The highest proportion of parents indicated they were very confident in knowing when their child needs to visit a healthcare provider (210, 35.2%) and in following their child’s diet or nutrition plan (220, 36.9%). This highlights the crucial role of parents in identifying the needs of their children, which include psychological, nutritional, and physical needs. In accordance, parents are usually the first to determine when their child needs to visit mental health services based on mental health issue signs [30,31,32]. An earlier analysis demonstrated increased or high levels of parental confidence in parenting a child with complex needs [33]. The literature declared that high parental confidence is correlated with better ability to handle various parenting challenges without increasing parental stress [18,34,35]. In contrast, parents of asthmatic children had poor confidence in determining or managing asthma exacerbations, and some of them were unsure about seeking medical assistance [36]. Low levels of parental confidence are reported to result in parental stress [37], consequently negatively impacting the child’s health and well-being [38]. These low levels of parental confidence and knowledge were due to multiple factors, including educational, lifestyle disturbances, prior negative experiences, insurance problems, or treatment costs [36]. Yet, empowering parents may aid in increasing parental knowledge, providing appropriate health services, and improving child health [39].

Regarding the finding that the highest proportion of parents indicated they were very confident in following their child’s diet or nutrition plan (220, 36.9%), it implies that most parents are proactive in managing and caring for their child and have high levels of self-efficacy. Previous studies revealed a positive association between parental self-efficacy, standardized body mass index for their child [40], and following a healthy nutrition plan for their child [40,41] (reducing intake of cake and cordial and increasing intake of vegetables, fruits, and water [42]). Indeed, the likelihood of children adhering to a healthy diet is influenced by their parents’ diet and parent–child communication. Previous investigations have highlighted that children of parents who follow a healthy diet also follow healthy diet habits [43] and that positive communication between parents and children is linked to children’s healthy nutrition [44,45,46]. Although there is a significant increase in parental confidence in following their child’s positive nutrition behaviors [47,48], the high risk of obesity and overweight among children worldwide [49,50,51,52,53] underscores the need to further increase this confidence.

The results of this study showed that “very confident” was the most selected response for knowing how to contact healthcare providers (229, 38.4%) and scheduling an appointment (221, 37.0%). In contrast, confidence was lower regarding setting up telemedicine visits, where the most common response was somewhat confident (201, 33.7%). Consistent with our findings, prior studies reveal that parents and patients prefer [54,55] or have more confidence in [56,57] in-person visits and medical appointments than telemedicine visits, despite the advantages of telemedicine. Telemedicine offers considerable benefits for children’s health services, but there is a lack of evidence regarding its safety and effectiveness [58], which may contribute to low confidence in telemedicine. Moreover, this confidence is linked to perceived self-efficacy [59]. An earlier investigation also documented several additional obstacles to using telemedicine services, like unfamiliarity, inadequate internet quality, and technological issues [60]. Hence, improving confidence in and acceptance of telemedicine is essential. Implementing practical control approaches and increasing skills for both parents and healthcare professionals may be beneficial [55,56].

Our results show that divorced parents were less likely to report a high level of parental self-efficacy compared to married parents (OR = 0.49, 95% CI: 0.26–0.92, *p* = 0.028). In line with this, prior studies have emphasized an association between marital or social support and parental self-efficacy [61,62]. Similarly, another earlier study has identified parents’ separations or divorce as an associated factor with lower parental self-efficacy [63]. These observations may partly be explained by strain, stress, anxiety, and depression among divorced parents [63,64]. Research indicates that the level of parental stress is associated negatively with parental self-efficacy [65]. Moreover, divorced or single parents reported challenges in decision-making when they lacked a partner to consult with [66,67,68]. Hence, programs aimed at increasing self-efficacy among divorced parents are necessary. Social support and other coping strategies may help enhance parental self-efficacy and improve child health and well-being [65,69].

In our study, higher income was strongly linked to greater self-efficacy, with parents earning 1001–1500 JOD showing significantly higher odds (OR = 4.44, 95% CI: 2.42–8.15, *p* < 0.001) compared to those earning less than 500 JOD. Additionally, having health insurance was associated with higher self-efficacy (OR = 0.50, 95% CI: 0.33–0.76, *p* = 0.001). Consistent with these findings, evidence from the literature underlines an association between parental self-efficacy and income; the higher the economic burden, the lower parents’ self-efficacy [70,71,72,73,74]. Several factors could explain these results. Firstly, there was reduced stress among higher-income parents and those having health insurance for their children. Parental stress is documented to be lower among higher-income parents and those having health insurance for their children than among lower-income parents [75,76,77] and those who do not have health insurance for their children [75]. Secondly, the likelihood of insurance coverage loss is higher among low-income parents due to changes in occupation status and household structure [78]. Additionally, the quality of healthcare differs for children with insurance and those without; children with insurance receive more healthcare services than uninsured children [79]. Thirdly, medication adherence is influenced by income status [80,81,82,83,84]. Finally, decision-making regarding childcare is also affected by parents’ income, reflecting the disparities in knowledge sources about childcare between high-income parents and low-income parents [85]. Therefore, improving self-efficacy for parents with low income and those who do not have health insurance for their children is required. However, research highlights challenges in enhancing medication management for children [86] and an expected increase in health inequalities based on a family’s economic status [87,88,89,90]. Accordingly, it is crucial to build efficacious approaches to address these challenges.

This study found that parents working in medical fields had higher odds (OR = 3.30, 95% CI: 1.69–6.45, *p* < 0.001) compared to those not working in any field. This finding is consistent with several previous investigations that have established an association between the absence of medical knowledge among parents and both a shortage of parental self-efficacy in medical care-related decision-making for their children and health troubles related to these decisions [91,92,93,94,95,96]. Thus, having adequate knowledge among parents is underscored as a significant factor in medical care-related decision-making, as it improves parental self-efficacy and understanding for choices during the decision-making process [68,91,97,98,99,100,101,102,103,104,105,106,107,108]. Other studies demonstrate that efficacious parenting approaches and understanding of child development are more elevated among parents with higher education levels [109,110,111,112,113]. Therefore, implementing education programs and activities may enhance parental self-efficacy and medical care-related decision-making for non-working parents.

The current investigation revealed that parents with 2–3 children (OR = 1.73, 95% CI: 1.00–3.00, *p* = 0.049) or 4–5 children (OR = 1.59, 95% CI: 1.05–3.63, *p* = 0.03) had greater odds of self-efficacy compared to those with one child. This result could be due to increased parental experiences when they have two or more children [114]. On the other hand, having more children may result in decreased self-efficacy as children’s needs increase and parental burden grows [114]. A prior investigation indicated that parents with two children experienced higher parental stress and lower self-efficacy compared to those with one child [115]. Indeed, these differences between studies may be attributed to disparities in culture [116,117], children’s health status [118], and parents’ age [118] and gender [119]. For instance, having more children is linked with resilience, emotional support, and happiness in collectivist cultures [116,117], such as Jordan. Ultimately, increased self-efficacy for parents with no children is imperative in Jordan.

This study has limitations. The use of convenience sampling via social media introduces selection bias and affects the generalizability of the study findings. Additionally, the cross-sectional survey study design restricted the ability to examine causality across the study variables [120]. Moreover, we did not have clinical data related to child health, which restricted the ability to examine the impact of parental self-efficacy on child health. Therefore, the study findings should be interpreted carefully. Future longitudinal studies that involve pediatric health data and examine the impact of health interventions on improving parental self-efficacy are recommended.

## 5. Conclusions

The majority of the parents in this study expressed strong self-efficacy in managing their child’s care, specifically in healthcare-related tasks. The study findings highlighted the importance of parental socioeconomic characteristics in shaping self-efficacy. In order to improve the health outcomes of their children and the self-efficacy of low-income families and parents who work outside the medical profession, additional financial, educational, and psychological support should be provided. Further research is needed to identify interventions that enhance parental self-efficacy for these high-risk sociodemographic groups.

## Figures and Tables

**Figure 1 healthcare-13-02280-f001:**
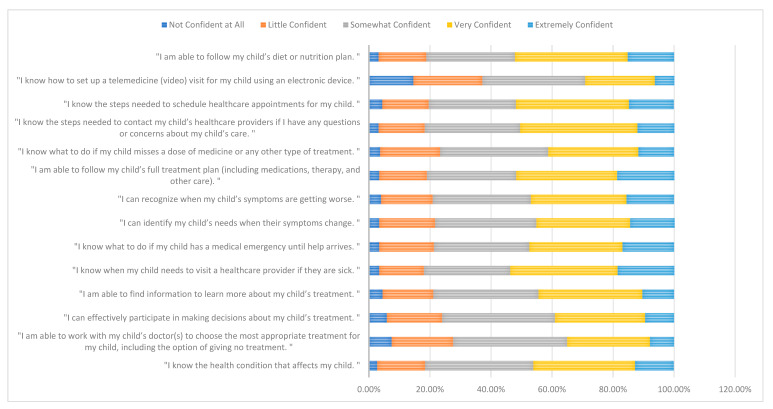
Self-efficacy of parents in managing their child’s medical care.

**Table 1 healthcare-13-02280-t001:** Sociodemographic characteristics of participants.

Sociodemographic Characteristics		N	%
Age (years)	18–23	41	6.9%
	24–29	84	14.1%
	30–34	133	22.3%
	35–39	79	13.2%
	40–44	102	17.1%
	45–49	77	12.9%
	50 and older	81	13.6%
Gender	Female	438	73.4%
	Male	159	26.6%
Marital status	Married	503	84.3%
	Widowed	31	5.2%
	Divorced	63	10.6%
Education level	Less than high school	116	19.4%
	Bachelor’s	269	45.1%
	Diploma	148	24.8%
	Postgraduate	64	10.7%
Income (JOD)	Less than 500	271	45.4%
	500–1000	209	35.0%
	1001–1500	91	15.2%
	1501 and above	26	4.4%
Occupation	Not working	239	40.0%
	Student	29	4.9%
	Working in medical fields	73	12.2%
	Working not in medical fields	211	35.3%
	Retired	45	7.5%
Insurance	No	186	31.2%
	Yes	411	68.8%
Comorbidities	No	424	71.0%
	Yes	173	29.0%
Number of children	1	105	17.6%
	2–3	246	41.2%
	4–5	174	29.1%
	6 and more	72	12.1%
Are you the one who takes care of the child?	No	87	14.6%
	Yes	324	54.3%
	Yes, alternatively with the other party where “the two parents share the responsibility and included in decision making”	186	31.2%

JOD: Jordanian dinar.

**Table 2 healthcare-13-02280-t002:** Parental self-efficacy score by sociodemographic characteristics.

Sociodemographic Variable		Mean ± SD	*p* Value
Age (years)	18–23	47.39 ± 11.89	0.18
	24–29	47.76 ± 11.51	
	30–34	47.27 ± 11.67	
	35–39	48.15 ± 11.23	
	40–44	45.30 ± 11.06	
	45–49	44.05 ± 10.85	
	50 and older	45.21 ± 13.37	
Gender	Female	46.93 ± 11.42	0.08
	Male	45.07 ± 12.30	
Marital status	Married	46.97 ± 11.37	0.01
	Widowed	45.61 ± 16.62	
	Divorced	42.59 ± 10.66	
Education level	Less than high school	41.92 ± 13.13	0.001
	Bachelor’s	49.03 ± 10.68	
	Diploma	43.42 ± 10.84	
	Postgraduate	50.69 ± 10.52	
Income (JOD)	Less than 500	43.10 ± 12.07	0.001
	500–1000	47.39 ± 11.02	
	1001–1500	52.54 ± 8.45	
	1501 and above	52.12 ± 10.62	
Occupation	Not working	45.00 ± 12.03	0.001
	Student	47.66 ± 13.30	
	Working in medical fields	53.33 ± 9.51	
	Working not in medical fields	45.99 ± 10.80	
	Retired	44.13 ± 12.25	
Insurance	No	46.85 ± 11.86	0.55
	Yes	46.25 ± 11.60	
Comorbidities	No	46.54 ± 11.20	0.73
	Yes	46.18 ± 12.80	
Number of children	1	46.19 ± 12.17	0.34
	2–3	47.41 ± 11.12	
	4–5	45.75 ± 12.05	
	6 and more	45.10 ± 11.89	
Are you the one who takes care of the child?	No	43.57 ± 14.65	0.04
	Yes	46.89 ± 10.81	
	Yes, alternatively with the other party	46.98 ± 11.45	

JOD: Jordanian dinar.

**Table 3 healthcare-13-02280-t003:** Factors associated with higher parental self-efficacy: logistic regression results.

Variable		OR (95%CI)	*p* Value
Age (years)	18–23	Reference	
	24–29	1.73 (0.69–4.34)	0.239
	30–34	1.10 (0.46–2.65)	0.827
	35–39	1.17 (0.45–2.99)	0.750
	40–44	0.78 (0.31–1.96)	0.599
	45–49	0.87 (0.33–2.29)	0.779
	50 and older	0.83 (0.30–2.28)	0.717
Gender	Female	Reference	
	Male	0.71 (0.43–1.18)	0.185
Marital status	Married	0.00 (0.00–0.00)	0.088
	Widowed	0.95 (0.42–2.16)	0.906
	Divorced	0.49 (0.26–0.92)	0.028
Education level	Less than high school	Reference	
	Bachelor’s	1.62 (0.93–2.82)	0.086
	Diploma	0.62 (0.35–1.11)	0.111
	Postgraduate	1.47 (0.69–3.13)	0.322
Income (JOD)	Less than 500	Reference	
	500–1000	1.51 (0.99–2.29)	0.055
	1001–1500	4.44 (2.42–8.15)	0.000
	1501 and above	1.82 (0.69–4.78)	0.224
Occupation	Not working	Reference	
	Student	1.08 (0.42–2.77)	0.872
	Working in medical fields	3.30 (1.69–6.45)	0.000
	Working not in medical fields	1.15 (0.72–1.85)	0.555
	Retired	0.61 (0.25–1.48)	0.271
Insurance	No	Reference	
	Yes	0.50 (0.33–0.76)	0.001
Comorbidities	No	Reference	
	Yes	0.74 (0.49–1.11)	0.149
Number of children	1	Reference	
	2–3	1.73 (1.00–3.00)	0.049
	4–5	1.95 (1.05–3.63)	0.035
	6 or more	2.23 (0.98–5.08)	0.057
Are you the one who takes care of the child?	No	Reference	
	Yes	1.10 (0.65–1.88)	0.723
	Yes, alternatively with the other party	1.37 (0.76–2.46)	0.301

JOD: Jordanian dinar.

## Data Availability

The original contributions presented in this study are included in the article. Further inquiries can be directed to the corresponding author.

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
