# Peer review of "Parental Self-Efficacy in Managing Pediatrics’ Medications and Treatments in Jordan: A Cross-Sectional Study"

_healthcare, 2025, doi:10.3390/healthcare13182280_

Round 1
Reviewer 1 Report
Comments and Suggestions for Authors
Thank you for the opportunity to review original research on parental self-efficacy in managing paediatrics’ medications and treatments in Jordan. I find the idea behind the research interesting and important, but there are several points which need to be addressed to ensure this research brings forward the best of its results.
In abstract and the manuscript ensure you use the same description for monetary values/currency; you have used “units”, “JOD”, “JD” and state what a certain abbreviation means.
“Parents with 2-3 children (OR = 1.73, 95% CI: 1.00– 3.00, p = 0.049) or 4-5 children (OR = 1.59, 95% CI: 1.05-3.63, p =0.03) had greater odds of self-efficacy compared to those with no child.” How can there be parents with no children? You do no have this in “number of children” stratification. Ensure the analysis and reporting is correct in results, discussion and abstract.
Introduction: Please further expand the introduction to focus on your research. I don’t find the sentence in line 64 and 65 important for this research. What are the coping strategies which promote health outcomes?
Methods: Merge paragraphs 2.2. and 2.5. and add a description based on what you calculated your sample size to be 383. In paragraph 2.3. Study instrument add minimum and maximum score. Was the questionnaire translated into Arabic or was it used in English? How was informed consent obtained if you used an online method of questionnaire collection? Did you collect information on the age of children? Could there be a difference in self-efficacy based on child’s age?
Why did you choose a cut-off point of 48?
Results: Regarding reported comorbidities; where they parents’ or children’s comorbidities? Parent taking care of a child with chronic illness can have a different set of skills and level of self-efficacy than parent of a “healthy” child. How was the question intended and could have the participants give a biased answer?
I find the term “Yes, alternatively with the other party” a bit clumsy. Does this mean two parents share a responsibility but the other parent in not included in decision making?
I would suggest you change table 2 for a graph or a figure of answer distribution for easier understanding, as the table is quite busy and very hard to comprehend all the results.
A lot of missing punctuation marks in paragraph in the lines 156-165.
Line 168: Divorced parents had lower odds of what?
Include statistics of how well the model fits the data
Discussion: please further expand or explain the connection between sentences in line 181-183 and 183-185. It seems that only mental health issues constitute health problems.
Explan how sentence in line 210-211 fits into your results? Were parents who are mother more likely to have problems in self-efficacy when it comes to diet?
Line 213: the results do not indicate they show the most common answer, please correct this.
Line 285: Parents with no children? Please explain
Comments on the Quality of English LanguageMinor grammar changes are needed
Author Response
Reviewer 1:
Thank you for the opportunity to review original research on parental self-efficacy in managing paediatrics’ medications and treatments in Jordan. I find the idea behind the research interesting and important, but there are several points which need to be addressed to ensure this research brings forward the best of its results.
In abstract and the manuscript ensure you use the same description for monetary values/currency; you have used “units”, “JOD”, “JD” and state what a certain abbreviation means.
- Thank you for this comment, we have now addressed this point throughout the manuscript.
“Parents with 2-3 children (OR = 1.73, 95% CI: 1.00– 3.00, p = 0.049) or 4-5 children (OR = 1.59, 95% CI: 1.05-3.63, p =0.03) had greater odds of self-efficacy compared to those with no child.” How can there be parents with no children? You do no have this in “number of children” stratification. Ensure the analysis and reporting is correct in results, discussion and abstract.
- Thank you for this comment, we have now addressed this point and corrected this typo mistake, see line 40 “it should be “one child””.
Introduction: Please further expand the introduction to focus on your research. I don’t find the sentence in line 64 and 65 important for this research. What are the coping strategies which promote health outcomes?
- Thank you for this comment, we have now addressed this point and removed the sentence highlighted in lines 70-72. Besides, we have now added the coping strategies which promote health outcomes, see lines 60-64.
Methods: Merge paragraphs 2.2. and 2.5. and add a description based on what you calculated your sample size to be 383.
- Thank you for this comment, we have now addressed this point and added sample size calculation equation, see lines 105-108.
In paragraph 2.3. Study instrument add minimum and maximum score. Was the questionnaire translated into Arabic or was it used in English? How was informed consent obtained if you used an online method of questionnaire collection? Did you collect information on the age of children? Could there be a difference in self-efficacy based on child’s age?
- Thank you for this comment, we have now addressed this point and added the minimum and maximum score (see line 120). The questionnaire translated into Arabic (see lines 111-112). Informed consent obtained was obtained through informing the study participants that completing the questionnaire tool is considered as an informed consent (see lines 147-148).
- Why did you choose a cut-off point of 48?
- Thank you for this comment, we have now addressed this point and corrected the typo and highlighted that we used 46.43 as a cut-off point as this was the mean self-efficacy score for the study sample. We used it to represent the central tendency of the self-efficacy of the study sample, see line 159.
Results: Regarding reported comorbidities; where they parents’ or children’s comorbidities? Parent taking care of a child with chronic illness can have a different set of skills and level of self-efficacy than parent of a “healthy” child. How was the question intended and could have the participants give a biased answer?
- Thank you for this comment. The whole questionnaire was directed towards the parents themselves. Therefore, all sociodemographic characteristics presented in Table 1 are related to the participants as highlighted in the title of the table. This was also, highlighted in the invitation letter of the questionnaire to avoid any biased answer.
I find the term “Yes, alternatively with the other party” a bit clumsy. Does this mean two parents share a responsibility but the other parent in not included in decision making?
- Thank you for this comment. Concerning this point, we meant that the two parents share a responsibility and included in decision making. We have now clarified this point in Table 1.
I would suggest you change table 2 for a graph or a figure of answer distribution for easier understanding, as the table is quite busy and very hard to comprehend all the results.
- Thank you for this comment, we have now addressed this point and replaced Table 2 with Figure 1.
A lot of missing punctuation marks in paragraph in the lines 156-165.
- Thank you for this comment, we have now addressed this point, see lines 200-209.
Line 168: Divorced parents had lower odds of what?
- Thank you for this comment. Divorced parents had lower odds of having better self-efficacy, see line 215.
Include statistics of how well the model fits the data
- Thank you for this comment, we have now addressed this point, see lines 212-214.
Discussion: please further expand or explain the connection between sentences in line 181-183 and 183-185. It seems that only mental health issues constitute health problems.
- Thank you for this comment, we have now addressed this point, see lines 229-235.
Explan how sentence in line 210-211 fits into your results? Were parents who are mother more likely to have problems in self-efficacy when it comes to diet?
- Thank you for this comment, we have now deleted this sentence to avoid any confusion, see lines 260-261.
Line 213: the results do not indicate they show the most common answer, please correct this.
- Thank you for this comment, we have now addressed this point, see line 263.
Line 285: Parents with no children? Please explain
- Thank you for this comment, we have now addressed this point and corrected this typo “one child”, see line 329.
Reviewer 2 Report
Comments and Suggestions for Authors
Thank you for the opportunity to review this interesting study.
Much of the work done in this study appears to have been conducted to a high standard, but I have some concerns.
The introduction sets out the background well.
The methods, require clarification:
What was the language of the original questionnaire? Did adapting it require translation to another language? If translation was required, the gold standard is for an independent forward and back translation to be made and correlation of the two versions. If this was done it should be stated, and if not, then state how the translation was performed. As a pilot study was made to validate the questionnaire, the method of translation is not critical to the validity of the study as presented.
More detail is required about the recruitment method. What were the selection criteria? They should be stated.
How exactly did the invitation work?
ow was consent collected? Most ethics committees will accept that completion of the online survey implies consent if this is clearly stated on screen before access to the survey proceeds. This usually also requires a statement that as participation is anonymous, withdrawal after submission is not possible.
On what platform was the questionnaire delivered? Most ethics committees require the use of a security compliant server for such surveys. Common platforms like Microsoft® Teams™ can be compliant with these requirements if the appropriate software settings are selected, but these details should be stated.
Data collected from Likert scales should not be considered as continuous variables - numbers in a Likert scale are category labels and should be treated as categorical data. Treating them as continuous variables assumes that all individuals perceptions of the difference between each category are the same interval, and this cannot be validated. Given the survey size this probably has not invalidated the results, but that this assumption was made, needs to be stated, or the results re-analysed using non parametric statistics. I think the simplest fix would be to just say that 'the selected category was given a numeric value corresponding to the category number.
The actual power calculation that was used to determine the survey size should be stated.
The results, given the assumption above, are clearly presented, apart from something missing in line 156
The total mean of the score was 46.43 11.67, with maximum and minimum of 70-14. appears to be missing ±.
The discussion is clear and appropriate.
The conclusion is appropriate.
With correction to the methods, I would welcome publication.
Author Response
Reviewer 2:
Thank you for the opportunity to review this interesting study.
Much of the work done in this study appears to have been conducted to a high standard, but I have some concerns.
The introduction sets out the background well.
The methods, require clarification:
What was the language of the original questionnaire? Did adapting it require translation to another language? If translation was required, the gold standard is for an independent forward and back translation to be made and correlation of the two versions. If this was done it should be stated, and if not, then state how the translation was performed. As a pilot study was made to validate the questionnaire, the method of translation is not critical to the validity of the study as presented.
- Thank you for this comment, we have now addressed this point and highlighted that we used the forward-backward translation to administer the questionnaire in the Arabic language, see lines 111-112.
More detail is required about the recruitment method. What were the selection criteria? They should be stated.
- Thank you for this comment, the inclusion criteria are mentioned in lines 97-98 “The study's participants were parents with children under the age of 18 who are currently living in Jordan.”. We have now added more details concerning the recruitment method, see lines 100-104.
How exactly did the invitation work?
- Thank you for this comment. We have now added more details concerning the recruitment method, see lines 100-104.
How was consent collected? Most ethics committees will accept that completion of the online survey implies consent if this is clearly stated on screen before access to the survey proceeds. This usually also requires a statement that as participation is anonymous, withdrawal after submission is not possible.
- Thank you for this comment. We have now added more details concerning informed consent, see lines 147-148.
On what platform was the questionnaire delivered? Most ethics committees require the use of a security compliant server for such surveys. Common platforms like Microsoft® Teams™ can be compliant with these requirements if the appropriate software settings are selected, but these details should be stated.
- Thank you for this comment. We have now added that we used QUALTRIC platform to deliver the questionnaire, see lines 100-101.
Data collected from Likert scales should not be considered as continuous variables - numbers in a Likert scale are category labels and should be treated as categorical data. Treating them as continuous variables assumes that all individuals perceptions of the difference between each category are the same interval, and this cannot be validated. Given the survey size this probably has not invalidated the results, but that this assumption was made, needs to be stated, or the results re-analysed using non parametric statistics. I think the simplest fix would be to just say that 'the selected category was given a numeric value corresponding to the category number.
- Thank you for this comment. We have now added the following sentence “The selected category was given a numeric value corresponding to the category number.” based on the reviewer’s comment, see lines 120-121.
The actual power calculation that was used to determine the survey size should be stated.
- Thank you for this comment. We have now addressed point, see lines 105-108
The results, given the assumption above, are clearly presented, apart from something missing in line 156
The total mean of the score was 46.43 11.67, with maximum and minimum of 70-14. appears to be missing ±.
- Thank you for this comment. We have now addressed this point, see line 200.
The discussion is clear and appropriate.
The conclusion is appropriate.
With correction to the methods, I would welcome publication.
- Thank you for your valuable comments.
Reviewer 3 Report
Comments and Suggestions for Authors
This manuscript (Parental Self-Efficacy in Managing Pediatrics’ Medications and Treatments in Jordan: A Cross-Sectional Study) addresses an important topic and has strengths in its structured approach and comprehensive data collection. However, several issues need attention before publication. Key among these are ensuring consistency and clarity in the presentation of results (especially the logistic regression findings), addressing grammatical and formatting errors, and providing a more critical discussion (including study limitations and implications). With these revisions, the paper will be much stronger.
Abstract
Strengths: The Abstract is structured with Background, Methods, Results, and Conclusion, giving a clear overview of the study’s purpose, design, and key findings. It reports the sample size (N=597) and major outcomes (high parental self-efficacy levels and associated factors). The use of percentages (e.g. “35.2% very confident”) and effect estimates (OR, 95% CI, p-values) provides quantitative insight.
Weaknesses: There are grammatical and clarity issues. For example, “Poor parent’s self-efficacy” should read “Poor parental self-efficacy”. The currency “units” is undefined (presumably Jordanian dinars). The Results section of the Abstract is overly detailed: it lists multiple ORs and CIs, which may overwhelm readers. Notably, the abstract text says “higher self-efficacy was significantly associated with parents’ socioeconomic characteristics” but then gives a contradictory OR for insurance (OR=0.50 suggests lower odds for insured parents). The phrase “OR = OR 0.50” appears to be a typographical error. The Conclusion ends abruptly and misses context or implications.
Suggestions: Revise for grammar and consistency (e.g. parents’ vs parental, specify “JOD” for currency). Simplify the Results summary by focusing on the most important findings, and consider moving detailed statistics (like multiple ORs) to the main text or tables. Clarify any contradictory statements (e.g. the interpretation of the insurance OR) or remove superfluous statistics. Ensure the Conclusion summarizes the main message clearly, possibly adding a brief implication (e.g. “This suggests that socioeconomic support may be needed to bolster parental confidence”).
Introduction
Strengths: The Introduction provides a reasonable definition of parental self-efficacy and explains its relevance to child health. It cites literature linking parental self-efficacy to child outcomes (e.g. adherence, stress, adaptation). The flow moves from broad concepts to the study’s aim, concluding with the statement that such studies are lacking in Jordan. This justifies the research gap and the study objective.
Weaknesses: Some citations do not clearly support the accompanying text. For instance, claims about coping strategies [6–11] and follow-ups [11] cite instrument or measure papers rather than evidence of effects. The narrative jumps between points (e.g. general parenting tasks, then severe conditions) without clear transitions. The focus is broad (general parenting, behavior) but only briefly ties back to medications or treatments. Context about Jordan’s healthcare environment or why this population is unique is minimal. Repetition occurs (e.g. effects of low self-efficacy are mentioned in several places).
Suggestions: Organize the background more cohesively. For example, after defining parental self-efficacy, discuss evidence from similar studies on medication management specifically (if any exist), or at least on health-related tasks. Ensure each claim is supported by an appropriate reference (e.g. use outcome studies rather than measure-development papers). Add a sentence on why Jordan is a pertinent setting (e.g. prevalence of chronic pediatric illness or medication practices) to strengthen the justification. The final paragraph should clearly state the study’s research question or hypothesis (beyond “aims to investigate”). Consider trimming repetitive points and ensuring terminology is consistent (e.g. always use “parental self-efficacy”).
Methods
Strengths: The Methods section describes an online cross-sectional survey with a defined timeframe (April–July 2025). Inclusion criteria (“parents with children under 18”) and recruitment (social media) are stated. The questionnaire is based on a previously validated 14-item scale [24], and domains of self-efficacy are listed. Content validity processes (expert panel, pilot) are reported. Sample size calculation and ethical approval are appropriately mentioned. The statistical analysis plan (descriptive, t-tests/ANOVA, logistic regression with median split) is detailed.
Weaknesses: The use of convenience sampling via social media introduces selection bias (likely more educated/urban respondents) and should be acknowledged. The instrument’s reliability (e.g. Cronbach’s alpha) in this sample is not reported. It is unclear if the questionnaire was administered in Arabic or needed translation – this should be specified. The description of logistic regression (“scores dichotomized based on the median at a cutoff of 48”) is somewhat abrupt; it would help to justify this choice. The Kolmogorov–Smirnov test is mentioned with a typo (“kolmogrov-smirnov”). There is no mention of how missing data were handled.
Suggestions: Clarify details of the instrument: indicate language (and if translated), and report its internal consistency reliability. Acknowledge limitations of the sampling method (non-probabilistic, internet-based) and discuss potential bias. Correct typos (e.g. “Kolmogorov–Smirnov”). Explain why self-efficacy was dichotomized at the median (perhaps referencing statistical conventions). If available, mention how many respondents (if any) had incomplete data and how these were treated. Optionally, cite reporting guidelines (e.g. STROBE for observational studies) to ensure completeness. Providing these details would improve transparency and reproducibility.
Results
Strengths: The results are comprehensive. Sociodemographic characteristics are clearly tabulated (Table 1) and summarized in text (e.g. age, gender, income). The responses to each self-efficacy item are presented (Table 2) with key findings highlighted (e.g. most parents “very confident” about contacting providers). Mean self-efficacy scores by subgroup are given (Table 3), and significant differences (e.g. by marital status, income) are noted with p-values. The logistic regression (Table 4) identifies predictors of high self-efficacy, matching the methods. Overall, the results follow logically from the analysis plan.
Weaknesses: There are minor inconsistencies and unclear statements. In the description of education (lines 441–442), the text appears garbled (“138 64 (10.7%) had a bachelor’s degree” is confusing and duplicates a category). The narrative sometimes repeats table values in text, which can be redundant. Crucially, the interpretation of some logistic results seems incorrect: the text claims insurance and employment in medical fields increase self-efficacy, but in Table 4 “Yes” insurance has OR=0.50 (<1), meaning insured parents had lower odds of high self-efficacy. Similarly, the text says “2-3 children or 4-5 children had greater odds compared to those with no child,” but the table’s reference is “1 child.” (Also, a parent with no children would not fit inclusion criteria.) Such discrepancies suggest errors in reference category reporting. Formatting-wise, ensure consistent abbreviations (e.g. JOD).
Suggestions: First, correct the descriptive errors (e.g. fix the education line, ensure percent labels are consistent). In text, focus on highlighting the most important results rather than enumerating every percentage (the tables suffice for detail). Crucially, reconcile the logistic regression interpretation: check coding of variables and ensure that the reference categories are correctly described (e.g. it should likely read “compared to those with one child”). Clarify the insurance finding: either correct the OR or adjust the wording to reflect the true direction. It may help to state the number of parents classified as “high” vs “low” self-efficacy after the median split. Where effect sizes are large (e.g. OR=4.44 for mid-income group), these should be emphasized. Include any reliability statistics for the scale here if obtained. In general, ensure that all text statements agree with the tables, and streamline the write-up to avoid redundancy.
Discussion
Strengths: The Discussion interprets each key finding and compares it with literature. The high overall self-efficacy is linked to better parenting and child outcomes (supported by refs [16,29,30]). The authors discuss specific items (nutrition, scheduling care, telemedicine) and cite studies (e.g. telemedicine confidence [50–55]). Socioeconomic findings (marital status, income, employment, insurance, number of children) are addressed with plausible explanations and relevant references. The authors acknowledge some contrasting evidence (e.g. lower confidence in asthma [31]) and suggest interventions (empowerment, education). This demonstrates comprehensive engagement with the results.
Weaknesses: The section is lengthy and at times unfocused, shifting rapidly among topics. The causal interpretation is too strong given the design: for example, saying divorced parents had “lower self-efficacy” implies a causal effect of divorce, whereas in this cross-sectional survey one can only note association (PMID: 27293245). No limitations of the study are explicitly discussed. The misinterpretation of odds (noted above) is echoed here (e.g. insurance “associated with higher self-efficacy” despite OR<1). Some points lack specificity: for instance, citing SciDaily for “Parents’ participation…Linked to Self-efficacy” [52] is inappropriate for an academic discussion; this should be replaced by the original study or omitted. The paragraph on nutrition and obesity goes into tangential detail (global obesity trends) that seems beyond the paper’s focus. There is no “Limitations” paragraph at all – e.g. no mention of selection bias, cross-sectional design, or lack of child health data – nor discussion of generalizability. Finally, the Discussion has minimal forward-looking commentary (no suggestions for future research or applications).
Suggestions: Reorganize the Discussion into clearer themes (e.g. overall findings, specific domains, sociodemographic factors). Add a dedicated paragraph on limitations: note that, as a one-time survey, causal relationships cannot be inferred (PMID: 27293245), and findings may not generalize beyond the sample. Acknowledge potential bias (e.g. internet sample, self-report). Correct the insurance interpretation: if insured parents truly had lower odds, say so and discuss possible reasons (or verify coding). Remove or replace non-peer references (e.g. [52]) with original studies. Trim the section on nutrition/obesity to only what directly relates to this study’s findings. Suggest future research directions (e.g. longitudinal studies, interventions to boost self-efficacy, exploring its effect on actual medication adherence). Throughout, avoid overstatements (use “associated with” rather than implying causality). On style, vary sentence structure and reduce redundancy to improve readability.
Conclusion
Strengths: The Conclusion succinctly restates the main findings: high parental self-efficacy overall, and its association with socioeconomic factors (marital status, income, employment, insurance, number of children). It correctly highlights that divorced parents exhibited lower confidence.
Weaknesses: The Conclusion is very brief and largely mirrors the abstract rather than providing new insight. It lacks mention of the study’s context or implications (e.g. what should be done with these findings). There is no reflection on limitations or cautions. It does not suggest future work or potential interventions. The language is somewhat flat (e.g. “significantly associated” is statistical jargon better in Results).
Suggestions: Expand slightly to emphasize the practical meaning of the results. For example: “These findings suggest that socioeconomically disadvantaged parents (e.g. lower income, divorced) may benefit from targeted support or education to improve confidence in managing their child’s care.” Acknowledge the limitation that this was a survey study. You might also point out a key strength (e.g. large sample) and a direction (e.g. testing interventions to raise self-efficacy). Remove statistical wording (like “significantly”). A well-rounded conclusion might read more: “…in managing their child’s health. These associations imply that interventions or policies aimed at financial assistance, insurance coverage, and family support could bolster parental confidence. Future research could investigate how to effectively enhance self-efficacy in these subgroups.” This would make the Conclusion more actionable.
References
Strengths: The reference list is extensive and largely relevant to parental self-efficacy, medication management, and related topics. Many recent studies and reviews are cited, which shows the authors engaged with current literature. The formatting generally follows MDPI style (numeric order, journal abbreviations).
Weaknesses: Some entries are not peer-reviewed sources. For example, reference [52] is a ScienceDaily news item (a popular press summary), which is not appropriate for a scholarly article. Reference [72] appears to be a government advisory (“US Surgeon General’s Advisory”), which may be acceptable but should be clearly formatted. There are minor inconsistencies: for example, some references include DOIs while others do not; a few lack complete author lists or are formatted oddly (e.g. [60] has “p. 0_1-266” which looks like a placeholder). The in-text citations and reference list should be double-checked to ensure every cited work is present and vice versa.
Suggestions: Remove or replace non-scholarly references (e.g. replace [52] with the original study if possible, or omit it). Verify each reference’s details (authors, title, journal, volume, pages) for accuracy and consistency. Include DOIs for all journal articles if required by the journal. Ensure reference formatting matches MDPI’s instructions (e.g. book titles in italics, consistent use of “&” vs “and”). Check that all in-text citation numbers correspond to the correct references. For clarity, list “Jordanian dinar (JOD)” in tables or text when first mentioned. If possible, streamline the reference list by removing any sources that are not directly cited in the text. A final pass to ensure uniformity (punctuation, abbreviation of journal names, year placement) would improve professionalism.
References for Methodological Points: Cross-sectional designs measure variables at one time point and cannot establish causality (PMID: 27293245); they are also subject to bias and reverse causation issues (PMID: 27293245). In line with existing literature, parental self-efficacy is influenced by socio-demographic context (e.g. SES) (DOI https://doi.org/10.1007/s40894-023-00216-w), which supports examining factors such as income and marital status in this study.
Author Response
Reviewer 3:
This manuscript (Parental Self-Efficacy in Managing Pediatrics’ Medications and Treatments in Jordan: A Cross-Sectional Study) addresses an important topic and has strengths in its structured approach and comprehensive data collection. However, several issues need attention before publication. Key among these are ensuring consistency and clarity in the presentation of results (especially the logistic regression findings), addressing grammatical and formatting errors, and providing a more critical discussion (including study limitations and implications). With these revisions, the paper will be much stronger.
- Thank you for your valuable comments, we have now addressed all comments as highlighted below.
Abstract
Strengths: The Abstract is structured with Background, Methods, Results, and Conclusion, giving a clear overview of the study’s purpose, design, and key findings. It reports the sample size (N=597) and major outcomes (high parental self-efficacy levels and associated factors). The use of percentages (e.g. “35.2% very confident”) and effect estimates (OR, 95% CI, p-values) provides quantitative insight.
Weaknesses: There are grammatical and clarity issues. For example, “Poor parent’s self-efficacy” should read “Poor parental self-efficacy”. The currency “units” is undefined (presumably Jordanian dinars). The Results section of the Abstract is overly detailed: it lists multiple ORs and CIs, which may overwhelm readers. Notably, the abstract text says “higher self-efficacy was significantly associated with parents’ socioeconomic characteristics” but then gives a contradictory OR for insurance (OR=0.50 suggests lower odds for insured parents). The phrase “OR = OR 0.50” appears to be a typographical error. The Conclusion ends abruptly and misses context or implications.
Suggestions: Revise for grammar and consistency (e.g. parents’ vs parental, specify “JOD” for currency). Simplify the Results summary by focusing on the most important findings, and consider moving detailed statistics (like multiple ORs) to the main text or tables. Clarify any contradictory statements (e.g. the interpretation of the insurance OR) or remove superfluous statistics. Ensure the Conclusion summarizes the main message clearly, possibly adding a brief implication (e.g. “This suggests that socioeconomic support may be needed to bolster parental confidence”).
- Thank you for this comment, we have now addressed all comments raised by the reviewer, see lines 18-48.
Introduction
Strengths: The Introduction provides a reasonable definition of parental self-efficacy and explains its relevance to child health. It cites literature linking parental self-efficacy to child outcomes (e.g. adherence, stress, adaptation). The flow moves from broad concepts to the study’s aim, concluding with the statement that such studies are lacking in Jordan. This justifies the research gap and the study objective.
Weaknesses: Some citations do not clearly support the accompanying text. For instance, claims about coping strategies [6–11] and follow-ups [11] cite instrument or measure papers rather than evidence of effects. The narrative jumps between points (e.g. general parenting tasks, then severe conditions) without clear transitions. The focus is broad (general parenting, behavior) but only briefly ties back to medications or treatments. Context about Jordan’s healthcare environment or why this population is unique is minimal. Repetition occurs (e.g. effects of low self-efficacy are mentioned in several places).
Suggestions: Organize the background more cohesively. For example, after defining parental self-efficacy, discuss evidence from similar studies on medication management specifically (if any exist), or at least on health-related tasks. Ensure each claim is supported by an appropriate reference (e.g. use outcome studies rather than measure-development papers). Add a sentence on why Jordan is a pertinent setting (e.g. prevalence of chronic pediatric illness or medication practices) to strengthen the justification. The final paragraph should clearly state the study’s research question or hypothesis (beyond “aims to investigate”). Consider trimming repetitive points and ensuring terminology is consistent (e.g. always use “parental self-efficacy”).
- Thank you for this comment, we have now polished the introduction section by adding better connection between sentences and evidence from previous study in Jordan, see lines 52-90.
Methods
Strengths: The Methods section describes an online cross-sectional survey with a defined timeframe (April–July 2025). Inclusion criteria (“parents with children under 18”) and recruitment (social media) are stated. The questionnaire is based on a previously validated 14-item scale [24], and domains of self-efficacy are listed. Content validity processes (expert panel, pilot) are reported. Sample size calculation and ethical approval are appropriately mentioned. The statistical analysis plan (descriptive, t-tests/ANOVA, logistic regression with median split) is detailed.
Weaknesses: The use of convenience sampling via social media introduces selection bias (likely more educated/urban respondents) and should be acknowledged. The instrument’s reliability (e.g. Cronbach’s alpha) in this sample is not reported. It is unclear if the questionnaire was administered in Arabic or needed translation – this should be specified. The description of logistic regression (“scores dichotomized based on the median at a cutoff of 48”) is somewhat abrupt; it would help to justify this choice. The Kolmogorov–Smirnov test is mentioned with a typo (“kolmogrov-smirnov”). There is no mention of how missing data were handled.
Suggestions: Clarify details of the instrument: indicate language (and if translated), and report its internal consistency reliability. Acknowledge limitations of the sampling method (non-probabilistic, internet-based) and discuss potential bias. Correct typos (e.g. “Kolmogorov–Smirnov”). Explain why self-efficacy was dichotomized at the median (perhaps referencing statistical conventions). If available, mention how many respondents (if any) had incomplete data and how these were treated. Optionally, cite reporting guidelines (e.g. STROBE for observational studies) to ensure completeness. Providing these details would improve transparency and reproducibility.
- Thank you for this comment. We have now clarified the language of the questionnaire tool (see lines 111-112). The internal consistency/reliability of the original questionnaire is not available; however, we have now added the findings related to the internal consistency across our study sample, which demonstrated a very high level of internal reliability “see lines 135-138”. We have now corrected the typo “see line 152”. We have now corrected the typo and justified the use of “mean” as a cut-off point for the logistic regression analysis. We did not have any missing data as completing the questionnaire items was compulsory to save the response. The limitations section was acknowledged in lines 338-346.
Results
Strengths: The results are comprehensive. Sociodemographic characteristics are clearly tabulated (Table 1) and summarized in text (e.g. age, gender, income). The responses to each self-efficacy item are presented (Table 2) with key findings highlighted (e.g. most parents “very confident” about contacting providers). Mean self-efficacy scores by subgroup are given (Table 3), and significant differences (e.g. by marital status, income) are noted with p-values. The logistic regression (Table 4) identifies predictors of high self-efficacy, matching the methods. Overall, the results follow logically from the analysis plan.
Weaknesses: There are minor inconsistencies and unclear statements. In the description of education (lines 441–442), the text appears garbled (“138 64 (10.7%) had a bachelor’s degree” is confusing and duplicates a category). The narrative sometimes repeats table values in text, which can be redundant. Crucially, the interpretation of some logistic results seems incorrect: the text claims insurance and employment in medical fields increase self-efficacy, but in Table 4 “Yes” insurance has OR=0.50 (<1), meaning insured parents had lower odds of high self-efficacy. Similarly, the text says “2-3 children or 4-5 children had greater odds compared to those with no child,” but the table’s reference is “1 child.” (Also, a parent with no children would not fit inclusion criteria.) Such discrepancies suggest errors in reference category reporting. Formatting-wise, ensure consistent abbreviations (e.g. JOD).
Suggestions: First, correct the descriptive errors (e.g. fix the education line, ensure percent labels are consistent). In text, focus on highlighting the most important results rather than enumerating every percentage (the tables suffice for detail). Crucially, reconcile the logistic regression interpretation: check coding of variables and ensure that the reference categories are correctly described (e.g. it should likely read “compared to those with one child”). Clarify the insurance finding: either correct the OR or adjust the wording to reflect the true direction. It may help to state the number of parents classified as “high” vs “low” self-efficacy after the median split. Where effect sizes are large (e.g. OR=4.44 for mid-income group), these should be emphasized. Include any reliability statistics for the scale here if obtained. In general, ensure that all text statements agree with the tables, and streamline the write-up to avoid redundancy.
- Thank you for this comment. We have now addressed all comments raised by the reviewer. The descriptive errors were fixed “see line 166”. The logistic regression findings were corrected to show that having health insurance was associated with parental self-efficacy. Besides, the findings concerning working in the medical field was accurate, see lines 219. The typo error in the following sentence ““2-3 children or 4-5 children had greater odds compared to those with no child” was corrected to be “one child” not “no child”, see lines 224.
Discussion
Strengths: The Discussion interprets each key finding and compares it with literature. The high overall self-efficacy is linked to better parenting and child outcomes (supported by refs [16,29,30]). The authors discuss specific items (nutrition, scheduling care, telemedicine) and cite studies (e.g. telemedicine confidence [50–55]). Socioeconomic findings (marital status, income, employment, insurance, number of children) are addressed with plausible explanations and relevant references. The authors acknowledge some contrasting evidence (e.g. lower confidence in asthma [31]) and suggest interventions (empowerment, education). This demonstrates comprehensive engagement with the results.
Weaknesses: The section is lengthy and at times unfocused, shifting rapidly among topics. The causal interpretation is too strong given the design: for example, saying divorced parents had “lower self-efficacy” implies a causal effect of divorce, whereas in this cross-sectional survey one can only note association (PMID: 27293245). No limitations of the study are explicitly discussed. The misinterpretation of odds (noted above) is echoed here (e.g. insurance “associated with higher self-efficacy” despite OR<1). Some points lack specificity: for instance, citing SciDaily for “Parents’ participation…Linked to Self-efficacy” [52] is inappropriate for an academic discussion; this should be replaced by the original study or omitted. The paragraph on nutrition and obesity goes into tangential detail (global obesity trends) that seems beyond the paper’s focus. There is no “Limitations” paragraph at all – e.g. no mention of selection bias, cross-sectional design, or lack of child health data – nor discussion of generalizability. Finally, the Discussion has minimal forward-looking commentary (no suggestions for future research or applications).
Suggestions: Reorganize the Discussion into clearer themes (e.g. overall findings, specific domains, sociodemographic factors). Add a dedicated paragraph on limitations: note that, as a one-time survey, causal relationships cannot be inferred (PMID: 27293245), and findings may not generalize beyond the sample. Acknowledge potential bias (e.g. internet sample, self-report). Correct the insurance interpretation: if insured parents truly had lower odds, say so and discuss possible reasons (or verify coding). Remove or replace non-peer references (e.g. [52]) with original studies. Trim the section on nutrition/obesity to only what directly relates to this study’s findings. Suggest future research directions (e.g. longitudinal studies, interventions to boost self-efficacy, exploring its effect on actual medication adherence). Throughout, avoid overstatements (use “associated with” rather than implying causality). On style, vary sentence structure and reduce redundancy to improve readability.
- Thank you for this comment. We have now addressed the reviewer’s comment and rephrased the sentence on lines 279-284 as the following “Our results show that divorced parents were less likely to report high level of parental self-efficacy compared to married parents (OR = 0.49, 95% CI: 0.26–0.92, p = 0.028). In line with this, prior studies have emphasized an association between marital or social support and parental self-efficacy”. The limitations have now been acknowledged at the end of the discussion “see lines 339-346”. The misinterpretation of odds has now been corrected. Concerning the point “Some points lack specificity: for instance, citing SciDaily for “Parents’ participation…Linked to Self-efficacy” [56] is inappropriate for an academic discussion; this should be replaced by the original study or omitted.”, we have now removed this sentence to avoid confusing the readers. Concerning the paragraph on nutrition and obesity all what we mentioned was “the high risk of obesity and overweight among children worldwide” to discuss our findings that “of parents indicated they were very confident in following their child's diet or nutrition plan”. The limitations and recommendations have now been added at the end of the discussion, see lines 338-346.
Conclusion
Strengths: The Conclusion succinctly restates the main findings: high parental self-efficacy overall, and its association with socioeconomic factors (marital status, income, employment, insurance, number of children). It correctly highlights that divorced parents exhibited lower confidence.
Weaknesses: The Conclusion is very brief and largely mirrors the abstract rather than providing new insight. It lacks mention of the study’s context or implications (e.g. what should be done with these findings). There is no reflection on limitations or cautions. It does not suggest future work or potential interventions. The language is somewhat flat (e.g. “significantly associated” is statistical jargon better in Results).
Suggestions: Expand slightly to emphasize the practical meaning of the results. For example: “These findings suggest that socioeconomically disadvantaged parents (e.g. lower income, divorced) may benefit from targeted support or education to improve confidence in managing their child’s care.” Acknowledge the limitation that this was a survey study. You might also point out a key strength (e.g. large sample) and a direction (e.g. testing interventions to raise self-efficacy). Remove statistical wording (like “significantly”). A well-rounded conclusion might read more: “…in managing their child’s health. These associations imply that interventions or policies aimed at financial assistance, insurance coverage, and family support could bolster parental confidence. Future research could investigate how to effectively enhance self-efficacy in these subgroups.” This would make the Conclusion more actionable.
- Thank you for this comment, we have now addressed these points in the conclusion, see lines 348-358.
References
Strengths: The reference list is extensive and largely relevant to parental self-efficacy, medication management, and related topics. Many recent studies and reviews are cited, which shows the authors engaged with current literature. The formatting generally follows MDPI style (numeric order, journal abbreviations).
Weaknesses: Some entries are not peer-reviewed sources. For example, reference [52] is a ScienceDaily news item (a popular press summary), which is not appropriate for a scholarly article. Reference [72] appears to be a government advisory (“US Surgeon General’s Advisory”), which may be acceptable but should be clearly formatted. There are minor inconsistencies: for example, some references include DOIs while others do not; a few lack complete author lists or are formatted oddly (e.g. [60] has “p. 0_1-266” which looks like a placeholder). The in-text citations and reference list should be double-checked to ensure every cited work is present and vice versa.
Suggestions: Remove or replace non-scholarly references (e.g. replace [52] with the original study if possible, or omit it). Verify each reference’s details (authors, title, journal, volume, pages) for accuracy and consistency. Include DOIs for all journal articles if required by the journal. Ensure reference formatting matches MDPI’s instructions (e.g. book titles in italics, consistent use of “&” vs “and”). Check that all in-text citation numbers correspond to the correct references. For clarity, list “Jordanian dinar (JOD)” in tables or text when first mentioned. If possible, streamline the reference list by removing any sources that are not directly cited in the text. A final pass to ensure uniformity (punctuation, abbreviation of journal names, year placement) would improve professionalism.
- Thank you for this comment. We have now removed non-scholarly reference. All references formatting will be checked by the production team once again after the manuscript acceptance.
References for Methodological Points: Cross-sectional designs measure variables at one time point and cannot establish causality (PMID: 27293245); they are also subject to bias and reverse causation issues (PMID: 27293245). In line with existing literature, parental self-efficacy is influenced by socio-demographic context (e.g. SES) (DOI https://doi.org/10.1007/s40894-023-00216-w), which supports examining factors such as income and marital status in this study.
- Thank you for this comment. We have now addressed these comments, see the limitations section (line 341) and lines 84-85.